# Gastroenteropancreatic Neuroendocrine Tumors—Current Status and Advances in Diagnostic Imaging

**DOI:** 10.3390/diagnostics13172741

**Published:** 2023-08-23

**Authors:** Daniel Vogele, Stefan A. Schmidt, Daniel Gnutzmann, Wolfgang M. Thaiss, Thomas J. Ettrich, Marko Kornmann, Meinrad Beer, Markus S. Juchems

**Affiliations:** 1Department of Diagnostic and Interventional Radiology, Ulm University Medical Center, Albert-Einstein-Allee 23, 89081 Ulm, Germany; stefan.schmidt@uniklinik-ulm.de (S.A.S.); wolfgang.thaiss@uniklinik-ulm.de (W.M.T.); meinrad.beer@uniklinik-ulm.de (M.B.); 2Department of Diagnostic and Interventional Radiology, Konstanz Hospital, Mainaustraße 35, 78464 Konstanz, Germany; daniel.gnutzmann@glkn.de (D.G.); markus.juchems@glkn.de (M.S.J.); 3Department of Nuclear Medicine, Ulm University Medical Center, Albert-Einstein-Allee 23, 89081 Ulm, Germany; 4Department of Internal Medicine I, Ulm University Medical Center, Albert-Einstein-Allee 23, 89081 Ulm, Germany; thomas.ettrich@uniklinik-ulm.de; 5i2SouI—Innovative Imaging in Surgical Oncology Ulm, University Hospital of Ulm, Albert-Einstein-Allee 23, 89081 Ulm, Germany; marko.kornmann@uniklinik-ulm.de; 6Department of General and Visceral Surgery, Ulm University Medical Center, Albert-Einstein-Allee 23, 89081 Ulm, Germany

**Keywords:** gastroenteropancreatic neuroendocrine neoplasia, magnetic resonance imaging, computed tomography, positron emission computed tomography, metastasis, sensitivity

## Abstract

Gastroenteropancreatic neuroendocrine neoplasia (GEP-NEN) is a heterogeneous and complex group of tumors that are often difficult to classify due to their heterogeneity and varying locations. As standard radiological methods, ultrasound, computed tomography (CT), magnetic resonance imaging (MRI), and positron emission tomography–computed tomography (PET/CT) are available for both localization and staging of NEN. Nuclear medical imaging methods with somatostatin analogs are of great importance since radioactively labeled receptor ligands make tumors visible with high sensitivity. CT and MRI have high detection rates for GEP-NEN and have been further improved by developments such as diffusion-weighted imaging. However, nuclear medical imaging methods are superior in detection, especially in gastrointestinal NEN. It is important for radiologists to be familiar with NEN, as it can occur ubiquitously in the abdomen and should be identified as such. Since GEP-NEN is predominantly hypervascularized, a biphasic examination technique is mandatory for contrast-enhanced cross-sectional imaging. PET/CT with somatostatin analogs should be used as the subsequent method.

## 1. Introduction

(Neuro)endocrine tumors are a heterogeneous and complex group of tumors that frequently express neural and endocrine markers. They arise from cells of the diffuse neuroendocrine system and may be localized in the intestine, pancreas, and bronchopulmonary system. When localized in the abdomen, previously, terms such as carcinoid and islet cell tumor were used synonymously for these tumors. Today, they are referred to as gastroenteropancreatic neuroendocrine neoplasms (GEP-NENs) [1].

GEP-NEN are rare tumors overall. Their incidence is about 2.5/100,000 per year in male patients and about 2.3/100,000 per year in females [2]. Studies published in recent years tend to show an increasing incidence for GEP-NEN [3,4]. However, this may be explained, at least in part, by the fact that recent improvements in cross-sectional imaging have made tumor detection easier, especially for smaller ones [5].

### 1.1. Tumor Classification

GEP-NENs are classified into several categories based on their size, location, and histopathological features [6,7]. The classification system most commonly used for GEP-NENs is the World Health Organization (WHO) classification system [8]. This system divides GEP-NENs into three categories based on their biological behavior and histological characteristics:-Well-differentiated neuroendocrine tumors (NETs): These tumors are low- to intermediate-grade and have a relatively indolent behavior. They are further classified based on their histological features as either typical or atypical carcinoids, or they are classified as NETs with features of either pancreatic endocrine neoplasms or gastrointestinal carcinoids.-Poorly differentiated neuroendocrine carcinomas (NECs): These tumors are high-grade and have an aggressive behavior. They are classified based on their histological features as either small-cell or large-cell NECs.-Mixed adenoneuroendocrine carcinomas (MANECs): These tumors are rare and contain both neuroendocrine and non-neuroendocrine components.

The grading of GEP-NENs is based on the Ki-67 proliferation index and the mitotic count (Table 1). The Ki-67 proliferation index represents the percentage of tumor cells that are actively dividing, while the mitotic count is the number of dividing cells microscopically observed per high-power field (hpf, historically) or, more accurately, by using the SI unit per mm^2^ [9]. The grade of a tumor is determined by the highest score of either the Ki-67 proliferation index or the mitotic count.

The TNM staging system for GEP-NEN is shown in Table 2, taking into account the size and location of the primary tumor, involvement of regional lymph nodes, and the presence of distant metastasis [10]. Based on the T, N, and M stages, the overall stage of GEP-NEN is determined (Table 3). The stage is a key factor in determining treatment options and predicting prognosis. Figure 1 gives a simplified overview of the approach to GEP-NENs according to Uri et al. [11].

### 1.2. Clinical Appearance, Symptoms, and Imaging Diagnostics

The symptoms of GEP-NENs can vary depending on their location, size, and aggressiveness [12]. They can be classified as either functionally active or non-active depending on whether or not they produce hormones causing symptoms. Most functionally active NENs are malignant or metastasize early. An exception are insulinomas, being benign and small in about 95% of cases. Functionally active tumors can cause symptoms such as diarrhea, flushing, abdominal pain, and weight loss, while functionally non-active tumors may not cause any noticeable symptoms until they become large enough to cause pain or obstruction. Table 4 shows common symptoms of GEP-NENs in relation to their location.

### 1.3. Imaging

For imaging, a multimodal diagnostic concept that includes multi-detector CT (MDCT), magnetic resonance tomography (MRI), and nuclear medicine examinations must be established. The value of the various imaging methods depends to a large extent on the type of tumor and its localization [5].

CT is the first diagnostic tool of choice in most cases, as it has good sensitivity and relatively good specificity and is readily available [13]. Despite its high sensitivity, it must be mentioned that CT may miss small pancreatic NENs or small gastrointestinal NENs. MRI, as an alternative modality, also has to deal with this limitation. For locating small insulinomas or gastrinomas, endosonography can also be of high value. Since 90% of NENs show somatostatin receptors, nuclear medicine examination methods are of great importance in the diagnosis of NENs [3,14]. Somatostatin receptors can be detected via somatostatin receptor scintigraphy and, more recently, by means of 68Gallium-DOTA-TOC/-NOC/-TATE positron emission tomography (PET) [15,16]. Nuclear medicine diagnostics are of therapeutic relevance because foci showing an uptake can be treated with biological therapy with somatostatin analogs and radioreceptor therapy with gallium- or lutetium-labeled somatostatin analogs [17,18]. An overview of the available diagnostic tools and new techniques is given in Table 5.

The different types of GEP-NENs, their typical appearances and symptoms, and the corresponding imaging capabilities are presented in the following.

## 2. Gastrointestinal NENs

NENs arise from neuroendocrine cells and can, therefore, occur at any location of the gastrointestinal tract [3]. In an analysis of databases from four European counties (Germany, France, Spain, and the UK), the small intestine was most commonly affected (56.5 to 63.6%). The large intestine with 15.8 to 24.9% and the stomach with 15.6 to 24.4% were less frequent [19]. In the study of Loosen et al., there was a predominance for the male sex [19]. The presentation of GEP-NENs of the gastrointestinal tract varies depending on localization, aggressiveness, tumor size, and hormonal function.

### 2.1. Stomach and Duodenum

In addition to the WHO grading described above, NENs of the stomach (gastric neuroendocrine neoplasms, gNENs) are usually divided into four types: Types I and II arise from the enterochromaffin cells of the gastrointestinal mucosa and submucosa [20]. Type I is by far the most common (70–80% of all gNENs) and presents as a small, often multifocal, and usually polyp-shaped tumor in the gastric corpus. There is an association with chronic gastritis type A. The tumors show a low proliferation (mostly G1/G2); they metastasize rarely (2–5%) and only in case of a tumor size greater than 2 cm [21]. Type II occurs in association with multiple endocrine neoplasia type I (MEN I). In addition, gNENs of type II are also low-proliferation tumors, but they metastasize more frequently [22]. Types III and IV are sporadic and unifocal. Type III is the second most common gNEN. Usually, these are large G2 tumors, and in up to 80% of cases, metastases are already present [23,24]. Type IV is the rarest gNEN with the poorest prognosis and up to 100% risk of metastasis. These tumors are highly proliferative and undifferentiated (G3) in the sense of neuroendocrine carcinomas [25].

Duodenal NENs account for approximately 2–3% of all GEP-NENs [26]. Tumors can be divided into gastrin-producing tumors, somatostatin-producing tumors, and functional inactive tumors, which are located roughly around the duodenal papilla [27,28,29]. Duodenal NENs are very often associated with endocrine syndromes. Approximately 90% of MEN I patients develop a gastrinoma, and 50% of patients with neurofibromatosis I develop a somatostatin-producing tumor. These juxtapapillary somatostatin-secreting tumors can lead to cholestasis or pancreatitis-like symptoms, which are the typical initial manifestations of these tumors. Approximately 50% of duodenal NENs are intraluminal and polypoid tumors.

### 2.2. Small Intestine

NENs of the small intestine account for approximately 17.3% of all NENs in North America; however, their incidence rates can differ regionally [30]. The ileum, especially the terminal ileum, is the most common localization of GEP-NENs of the small intestine; the jejunum is less frequently affected (Figure 2). They produce serotonin and are often scattered in the terminal ileum [31]. In contrast to NENs of the duodenum, NENs in the rest of the small intestine are relatively aggressive tumors. At the time of diagnosis, there are usually metastases in lymph nodes and in about 20% of cases in the liver. This can lead to a carcinoid syndrome with diarrhea and flushing because serotonin is degraded in the liver and cannot enter the systemic circulation until liver metastasis has occurred. A distinctive feature of ileal NENs is a desmoplastic fibrous reaction of the surrounding mesenteric adipose tissue, which may involve mesenteric vessels and intestinal loops [32]. The consequences of mesenteric involvement can be diverse. On the one hand, there may be bridal distortion; on the other hand, there may even be signs of intestinal ischemia with vascular involvement (vascular tortuosity). Patients with ileal NENs without hepatic metastases have nonspecific symptoms, such as malaise or irritable bowel syndrome [33].

### 2.3. Appendix and Colon/Rectum

NENs of the appendix are usually clinically silent, serotonin-producing tumors. Patients with appendiceal NENs are usually younger than patients with NENs in other locations and account for 0.16–2.3% of all appendectomies [34]. NENs of the appendix are often seen as incidental findings during appendectomies, which are more frequently performed in young patients. Malignant degeneration is found in about 10% of cases and metastases are extremely rare [2,35]. Tumor size is considered to be the most reliable indicator of the malignant potential of appendiceal NENs [36]. Therefore, these tumors can usually be removed curatively in the course of an appendectomy.

While NENs of the colon represent an absolute rarity, NENs of the rectum have an estimated annual incidence of 1.04 per 100,000 persons. They usually manifest as small (<1 cm), hormonally inactive tumors that are displaced above the muscle layer and can be ablated endoscopically [37]. Rectal NENs account for approximately 25% of gastrointestinal NENs, are clinically inapparent in about half of cases, and are often found during routine rectal examinations. In the vast majority of cases, they are benign tumors. Larger tumors may be evident due to rectal bleeding, pain, or constipation [38].

### 2.4. Imaging

CT and MRI have different diagnostic values depending on the location and size of a digestive NEN. Particularly, gastric, colon, and rectal tumors are, at best, complementary to endoscopic procedures [39,40]. In the diagnosis of intestinal NENs, radiological cross-sectional imaging and nuclear medicine examinations for type I and type II tumors are of minor importance compared to endoscopic procedures [41]. In the case of larger type II tumors, radiological cross-sectional imaging can be considered as an alternative diagnostic method. For the more aggressive type III and type IV tumors, CT and MRI are used for tumor staging and, especially, to exclude metastases [42]. NENs of the duodenum are preferably diagnosed via endoscopy/endosonography; CT and MRI play a minor role in primary diagnostics. If a cross-sectional examination is nevertheless performed, gastrointestinal distension with an orally administered contrast medium is recommended prior to the examination, e.g., water for CT or T2w-negative contrast agents, such as manganese-containing fruit juices (e.g., pineapple juice), for MRI. In addition, i.v. contrast media should be used [43,44]. Duodenal NENs usually present as hypervascularized tumors. On the other hand, cross-sectional imaging procedures are important for local and distant staging. At the time of diagnosis of duodenal NENs, locoregional lymph node metastases are often found, whereas liver metastases occur rather late in the course of the disease [45]. To improve the local diagnosis, both CT and MRI should be performed in enterography [5,46]. The morphological spectrum of ileal GEP-NENs ranges from intraluminal or submucosal nodular lesions to stenosing bowel wall thickening with infiltration of surrounding mesenteric fatty tissue. Typically, this tumor entity shows severe hypervascularization in the arterial contrast phase [47]. CT enteroclysis achieves a sensitivity of 84.7% and a specificity of 96.9% for tumors up to 3 cm [48]. In a recent study by Morani et al., CT scans with negative intraluminal contrast were significantly more sensitive for concordant results than CT scans with positive intraluminal contrast. However, the specificity was not significantly different for negative versus positive enteric contrast (100% versus 93%) [46]. According to recent data, CT enterography offers comparable detection rates and achieves a per-patient sensitivity of 86%, a specificity of 100%, and an accuracy of 89% [49]. MR enterography has an even slightly higher sensitivity of approximately 95% [50]. In appendiceal NENs, imaging plays a role in detecting metastases. Although there have been advances in CT and MR diagnostics of the colon and rectum, the diagnosis of colonic and rectal NENs remains the domain of endoscopy and endosonography. However, it should be mentioned that cross-sectional imaging is also of great importance in the diagnosis of local spread and in the detection of metastases.

## 3. Pancreatic NENs

Pancreatic NENs are relatively rare, accounting for only a small percentage of pancreatic tumors, with an estimated annual incidence of 0.7 per 100,000 persons [51]. The two most common tumors, insulinoma and gastrinoma, have an annual incidence of 0.3–3 per 1 million, while the remaining tumor types are much less common [52]. Diagnosis of these tumors is usually based on a combination of characteristic clinical symptoms and laboratory findings. Imaging is used to locate the tumor and detect metastases (Figure 3). A distinction is made between hormone-active and non-hormone-active tumors. Hormone-active tumors exhibit a variety of clinical symptoms that are specific to the hormones they produce.

### 3.1. Insulinoma

Insulinoma is the most common endocrine pancreatic tumor and accounts for about 50% in this group [53]. It occurs between the ages of 30 and 60 and is more or less equally prevalent in both sexes. The hypoglycemia due to increased insulin secretion causes symptoms such as weakness, short-term unconsciousness, sweating, tremors, palpitations, and seizures accompanied by tachycardia [54]. Most of these tumors (90%) are smaller than 2 cm and grow in a solitary manner. They are usually benign in nature; however, 6–10% of insulinomas are malignant and show infiltration into the surrounding tissue, lymph node involvement, or liver metastases [55]. Multifocal tumors account for about 10% of cases. They are particularly found in the context of MEN1 syndrome (insulinoma, parathyroid adenoma, and pituitary adenoma) [56].

### 3.2. Gastrinoma

Gastrinomas are the second most common tumor group and account for about 20% of cases [53]. The majority of cases (about 60%) are malignancies [57]. Patients are usually around 40 years of age, and male sex is slightly preferred [58]. Increased gastrin secretion leads to Zollinger–Ellison syndrome with gastric ulceration and diarrhea. Between 85 and 90% of tumors are located in the gastrinoma triangle [59]. This term describes the triangular structure including the pancreatic head, duodenal C, and inflow of the ductus cysticus into the bile duct. Gastrinomas of the pancreas are more frequently malignant than gastrinomas of the duodenal wall [41]. Approximately 20–25% of gastrinomas are associated with MEN I. Because these tumors often are multiple and extrapancreatic, they are more difficult to localize than insulinomas [56].

### 3.3. Other Endocrine Neoplasms of the Pancreas

For completeness, the very rare VIPomas, glucagonomas, and somatostatinomas should also be mentioned. VIPomas are located intrapancreatically in up to 90% of cases and are malignant in 50–75%. Clinical symptoms are expressed in the Verner–Morrison syndrome (diarrhea, electrolyte disturbances, hyperglycemia, flush) [60]. The glucagonoma, which is always localized intrapancreatically, rarely produces hormones and is usually large and malignant [61]. Tumors cause glucagon syndrome, in which plasma glucagon levels are 10 to 1000 times higher than normal. Clinically, they are manifested, for example, by diabetes mellitus, necrotizing and migratory eczema (erythema necrolyticum migrans), anemia, and weight loss. Somatostatinomas occur in 35% of cases in the pancreas. They have a risk of malignancy of about 70% and are clinically manifested by diabetes mellitus, diarrhea, gallstones, and steatorrhea.

#### Hormone-Inactive Neuroendocrine Neoplasms of the Pancreas

The prevalence of hormone-inactive tumors ranges from 14% to 30% [53]. These tumors usually cause symptoms such as pain, jaundice, intestinal obstruction, or weight loss due to their size and location and, thus, resemble ductal adenocarcinoma. In most cases, they are slow-growing, often rather large at diagnosis, and usually malignant. Similarly, to hormone-active tumors of the pancreas, malignant degeneration manifests via lymph node involvement, infiltration into the surrounding tissues, and distant metastases [62].

### 3.4. Imaging

Both CT and MRI have a high accuracy in the diagnosis of pancreatic NENs. Multidetector spiral CT (MDCT) is, after sonography, the most frequently used method for their detection and localization [62]. The sensitivity is up to 94% in relation to tumor size and vascularization. Usually, endocrine tumors strongly absorb contrast media and are well-demarcated, at least in the arterial phase. Some tumors do not show noticeable contrast enhancement until the later phase. Small or atypical tumors are associated with high tumor grade and poor prognosis appear homogeneous, and larger ones may appear cystic and inhomogeneous due to central necrosis (Figure 4) [63,64]. Large tumors (>5 cm) are frequently malignant. In addition, calcifications in tumors are considered to be indicative of malignancy in hormonally inactive tumors [65]. Affected lymph nodes are also frequently hypervascularized. Compared with the surrounding pancreatic parenchyma, NENs of the pancreas in MRI typically show a hypointense signal in T1w and an increased signal in T2w sequences [66]. A study by Owen found that 48.3% of tumors showed such signaling behavior, and only 3.4% showed a T1w hyper- and T2w hypointense signal [67]. The tumors are usually severely hypervascularized and take up gadolinium in a homogeneous, annular, or heterogeneous manner [68]. Some rare cases with cystic degeneration only show contrast enhancement in the marginal area. The relationship of the location of the tumor to the pancreatic duct can be determined by high-resolution magnetic resonance cholangiopancreaticography (MRCP), where NENs of the pancreas rarely obstruct the pancreatic duct and blood vessels [69]. Some studies have also shown an additional value via DWI (diffusion-weighted imaging), especially in the differentiation between malignant and non-malignant pancreatic NENs [70] and for detecting lymph node metastases [71].

## 4. Nuclear Medicine Imaging Techniques

In terms of both pancreatic NEN imaging and gastrointestinal NEN imaging, functional nuclear medicine examination methods are becoming increasingly important (Figure 5).

In addition to the classic cross-sectional imaging described above, nuclear medicine takes advantage of the unique property of neuroendocrine tumors of expressing somatostatin receptors (SSTRs). GEP-NENs preferentially express subtype 2 of SSTRs. These receptors represent the target structure for nuclear medicine diagnostics, as they can be visualized with radiolabeled somatostatin analogs (SSAs, mainly 68Ga-DOTATOC and 68Ga-DOTATATE). The development of SSA DOTATOC, DOTANOC, and DOTATATE with high receptor affinity for SSTR 2 and their labeling with 68Gallium enabled the detection and localization of neuroendocrine tumors by using PET/CT or PET/MRI with high pooled sensitivity and specificity of 93% and 96% in NETs in a meta-analysis [72], and the detection rates of SSTR-PET were recently reviewed elsewhere [73]. However, with increasing tumor dedifferentiation, fewer SSTRs can be expressed. Thus, the use of SSTR-PET is proposed to be limited for G3 tumors [74], and the use of FDG PET/CT is suggested. However, in the study of You et al., who investigated the role of DOTATATE PET/CT in patients with low-/intermediate- versus high-grade NENs, all patients with high-grade NENs had positive DOTATATE PET/CT. SSTR-PET is also used to evaluate potential peptide receptor radiotherapy (PRRT) [75]. In pancreatic NENs, nuclear medicine techniques can achieve slightly higher sensitivities compared with CT and MRI (Figure 6) [76].

Schmidt-Tannwald et al. were able to show the superiority of 68Ga-DOTATE-PET/CT over MRI, including DW-MRI [77]. Although this study showed that diffusion imaging adds value compared to standard MRI, MRI was inferior to PET/CT even when DWI was used. PET/CT achieved a sensitivity of 100%, whereas MRI including DWI achieved only 64%, a value that seems to be quite low. The study of Sadowski et al. analyzed the added value of 68Ga-DOTATE-PET/CT with respect to all GEP-NEN localizations [78]. This study, which included 131 patients, investigated the GEP-NEN detection rates of functional nuclear medicine techniques (68Ga-DOTATE-PETCT and 111In pentetreotide SPECTCT and anatomic imaging (CT or MRI)) in patients whose primary tumor localization was unknown. 68GaDOTATE PET/CT showed the highest sensitivity, detecting 95.1%, and it was more sensitive than the other modalities for all organs (pancreas, small intestine, bone, etc.), except for the lungs and mediastinum.

### PET/MRI

With the introduction of PET/MRI systems, we are able to combine the analysis of SSTR-PET and MRI in NET. While it is well known that both SSTR-PET/CT and whole-body MRI show advantages and disadvantages in certain body areas [79], the combined analysis shows advantages regarding several aspects [80]—in particular, in overcoming potential discrepancies between lesion detection on MRI and SST-PET [81] and as a “one-stop shop” modality [82]. Figure 7 shows an example of a patient with a pancreatic NEN undergoing PET/MRI imaging. Prognostic factures derived from the combination of MRI and PET values remain under investigation, especially when considering ADC values to identify higher-grade lesions [83,84] or monitoring of treatment that is potentially combined with radiomics [84,85].

## 5. Radiomics

Unlike previous research, which focused on visual assessment and interpretation of radiological images, radiomic features represent the conversion of quantitative image features into datasets that are invisible to the human eye. Radiomic features are both shape features of a defined region of interest (ROI), such as volume or sphericity, and higher-order features that reflect the distribution of voxel values or the arrangement of different voxel intensities in the ROI [86]. Especially in the field of oncology, radiomics is comparatively well developed. Quantitative features of tumor tissues have great potential to provide access to information about the tumor phenotype that is not detectable by other means [87]. Radiomics can also be applied to GEP-NENs during diagnosis and staging. With a radiomics analysis based on contrast-enhanced computed tomography (CECT), Chiti et al. developed a tumor-grade model for GEP-NENs [88]. In addition, radiomics-based nomograms provide the possibility for preoperatively predicting tumor grading in patients with pancreatic neuroendocrine tumors (PNETs) [89,90]. The studies available to date show encouraging results with regard to the radiomic features of GEP-NENs. Further investigations are still necessary until these radiomic features can be used in the clinical routine.

## 6. Conclusions

GEP-NENs are a heterogeneous group of complex tumors. Since GEP-NENs are rare overall, it is important for radiologists to appreciate this tumor entity, which can be located ubiquitously in the abdomen. In the case of hormone-active tumors, patients usually initially present with typical symptoms, and the radiologist must ultimately identify the primary tumor location, perform diagnostics of the surroundings, and exclude metastases. According to the revised WHO classification of 2010, tumors are divided into three grades. Classification can be difficult due to heterogeneity and different localizations. CT and MRI have high detection rates for GEP-NENs and should be involved in the primary workup. In addition, nuclear medicine procedures such as PET/CT and PET/MRI show the highest sensitivity. In the future, detection rates and grading could be improved by technical advancements, such as radiomic features.

## Figures and Tables

**Figure 1 diagnostics-13-02741-f001:**
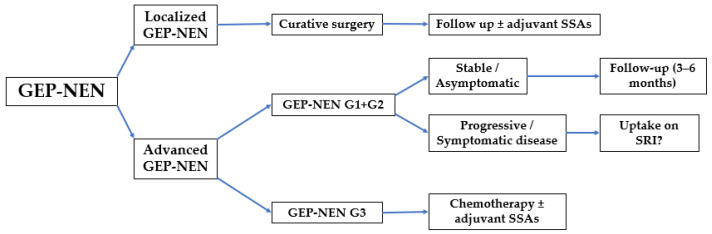
Flow diagram for the approach to GEP-NENs. NEN, neuroendocrine neoplasm; G; grade; SSA, somatostatin analog; SRI, somatostatin receptor imaging.

**Figure 2 diagnostics-13-02741-f002:**
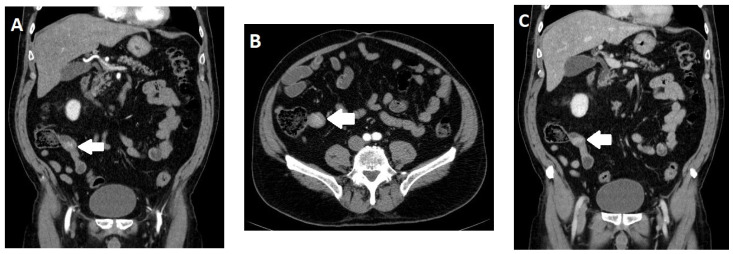
Computed tomography (CT) of a 65-year-old male patient with a neuroendocrine neoplasm (NEN) of the terminal ileum. In the arterial phase, a slightly hypervascularized tumor can be seen in the coronal (**A**) and axial (**B**) reconstructions (arrows). In the venous phase (**C**), the tumor shows homogenous enhancement (arrow).

**Figure 3 diagnostics-13-02741-f003:**
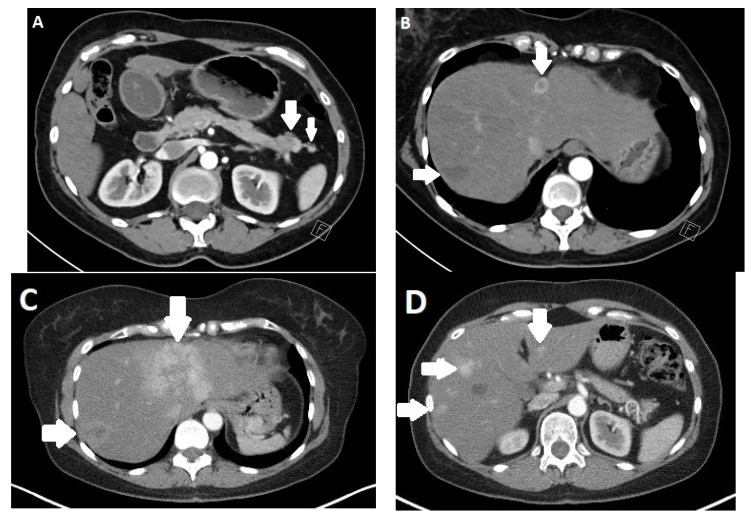
Computed tomography (CT) of a 52-year-old female patient with a neuroendocrine neoplasm (NEN) of the pancreatic tail. (**A**) shows a partially hypervascularized tumor and an adjacent satellite node (arrows). Near the falciforme ligament, a hypervascularized metastasis can be seen (**B**). Follow-up CT after 3 months (**C**,**D**) shows progressive hypervascularized liver metastasis (arrows).

**Figure 4 diagnostics-13-02741-f004:**
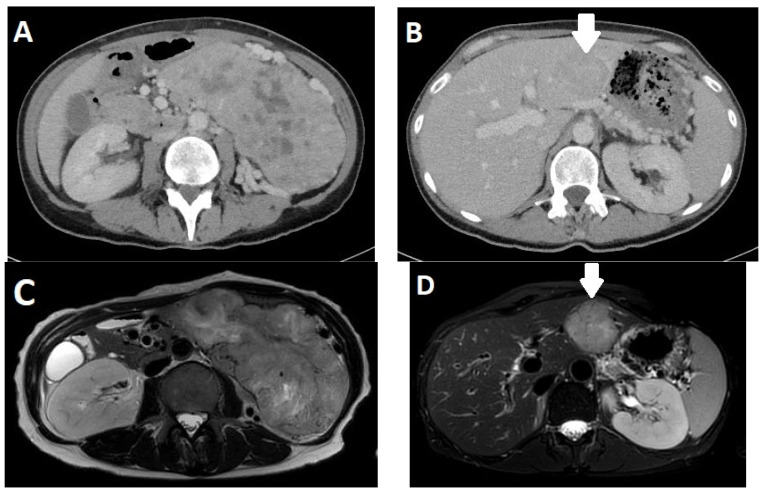
A 50-year-old female patient with a neuroendocrine neoplasm (NEN) of the pancreas. Computed tomography (CT) in the venous phase (**A**) shows a large inhomogeneous tumor in the left abdomen with hypodense portions due to necrosis. (**B**) shows a hypointense metastasis in liver segment II/II (arrow). In the corresponding magnetic resonance imaging (MRI) with T2-weighted sequences without (**C**) and with (**D**) fat suppression, the tumor and liver metastasis appear hyperintense (arrow).

**Figure 5 diagnostics-13-02741-f005:**
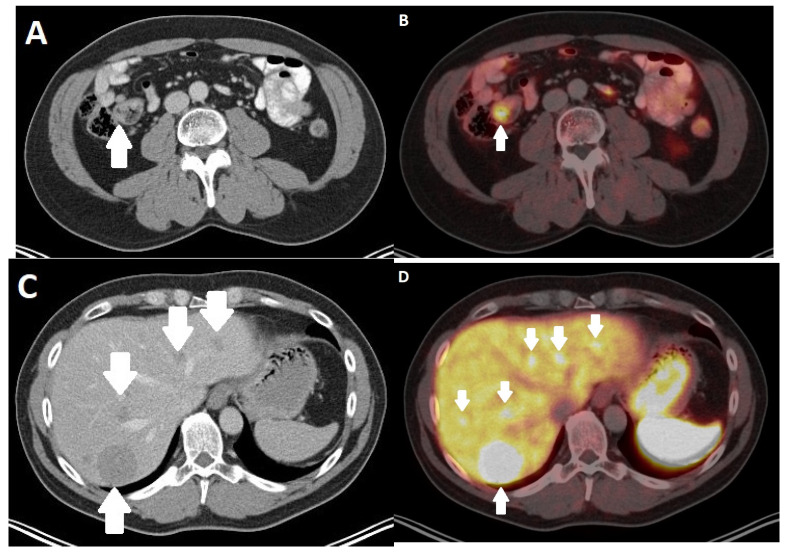
Positron emission tomography/computed tomography (PET/CT) of a 60-year-old male patient with a neuroendocrine neoplasm (NEN) of the terminal ileum. In CT, a small tumor can be seen in the axial (**A**) reconstructions (arrow). In PET fusion (**B**), the tumor shows expression of SSTR (arrow). In addition, hypodense liver metastases (arrows) can be seen (**C**) with the greatest lesion in segment VII (arrowhead). In PET fusion (**D**), the liver metastases show heterogenous expression of SSTR (arrows).

**Figure 6 diagnostics-13-02741-f006:**
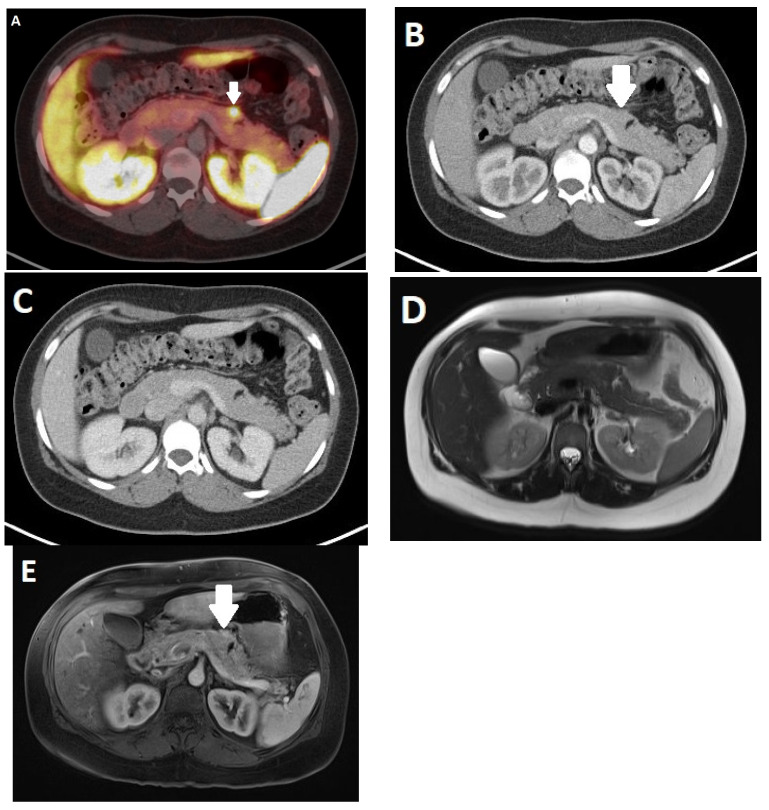
Positron emission tomography/computed tomography (PET/CT) of a 31-year-old female patient with MEN1 mutation and four SSTR-expressing pancreatic lesions who was under suspicion for neuroendocrine neoplasms (NENs). (**A**) One of the SSTR-expressing lesions (arrow) in the pancreatic body/tail adjacent to a small fatty lesion. The lesion is slightly hyperdense in the arterial phase (**B**) and is not visible in the venous phase (**C**) (arrows). The corresponding MRI shows no signal alteration in axial T2-weighted images (**D**) and only slight enhancement (arrow) in T1w after contrast admission (**E**).

**Figure 7 diagnostics-13-02741-f007:**
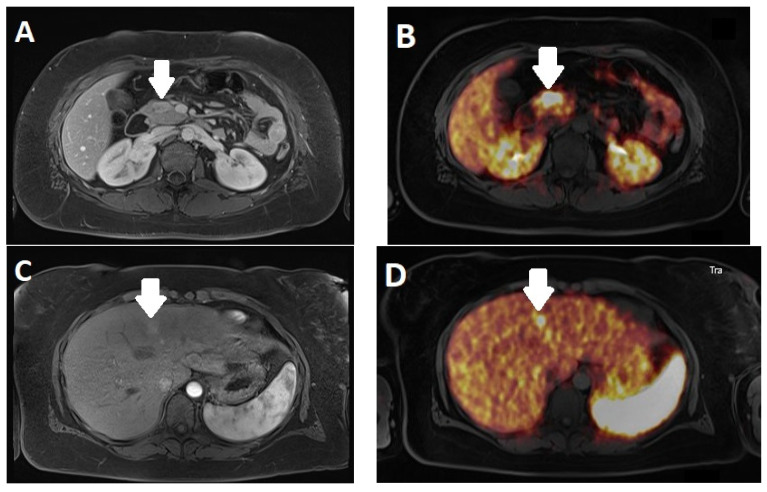
Positron emission tomography/magnetic resonance imaging (PET/MRI) of a 27-year-old female patient with a neuroendocrine tumor of the pancreas. MRI shows an enhancing lesion in the pancreatic head (**A**) with corresponding SSTR expression (**B**) (arrows). In segment IVa of the liver, an ill-defined enhancing lesion (**C**) can be seen in the arterial phase (arrows) with the corresponding SSTR expression (**D**). The lesions are both hard to detect in computed tomography (CT) in both the venous (**E**,**F**) (arrows) and arterial phase (not shown).

**Table 1 diagnostics-13-02741-t001:** WHO classification for GEP-NENs.

WHO-Grade	Ki-67 Index	Mitotic Count/10 HPF
Grade 1 NET	<3%	10
Grade 2 NET	3–20%	2–20
Grade 3 NET	>20%	>20
Grade 3 NEC	>20%	>20

HPF, high-power field; GEP-NEN, gastroenteropancretic neuroendocrine neoplasm; NET, neuroendocrine tumor; NEC, neuroendocrine carcinoma.

**Table 2 diagnostics-13-02741-t002:** TNM classification for GEP-NENs. The T stage is exemplary for neuroendocrine gastric tumors.

T Stage	Description
TX	Primary tumor cannot be assessed
T0	No evidence of primary tumor
T1	Tumor limited to the submucosa or muscularis propria
T2	Tumor invades beyond the muscularis propria into the subserosa or serosa without invasion of adjacent structures
T3	Tumor invades adjacent structures
T4	Tumor invades through the wall of the visceral peritoneum or into adjacent organs or structures
**N stage**	**Description**
NX	Regional lymph nodes cannot be assessed
N0	No regional lymph node metastasis
N1	Regional lymph node metastasis
**M stage**	**Description**
MX	Distant metastasis cannot be assessed
M0	No distant metastasis
M1	Distant metastasis

**Table 3 diagnostics-13-02741-t003:** Different stages of GEP-NEN.

Stage	Description
I	Small tumors that have invaded nearby tissues but have not spread to lymph nodes or distant sites.
II	Larger tumors that have invaded nearby tissues and may or may not have spread to nearby lymph nodes but have not yet metastasized to distant sites.
III	Tumors of any size that have spread to nearby lymph nodes but have not yet metastasized to distant sites.
IV	Tumors of any size that have metastasized to distant sites in the body, such as the liver, lungs, or bones.

GEP-NEN, gastroenteropancretic neuroendocrine neoplasm.

**Table 4 diagnostics-13-02741-t004:** Common symptoms of GEP-NENs.

Symptom	Description
Abdominal pain	Patients with GEP-NEN may experience persistent abdominal pain, especially in the upper abdomen.
Diarrhea or constipation	Changes in bowel habits, including diarrhea or constipation, can occur due to the presence of a tumor in the gastrointestinal tract.
Gastrointestinal bleeding	GEP-NEN tumors may cause bleeding in the gastrointestinal tract, resulting in hematochezia or hematemesis.
Heartburn or acid reflux	Patients with tumors in the stomach or esophagus may experience symptoms of acid reflux or heartburn.
Unintentional weight loss	Patients with GEP-NEN may lose weight without intending to due to a lack of appetite or malabsorption.
Jaundice	Tumors in the pancreas or bile ducts may cause jaundice, which is characterized by yellowing of the skin and eyes.
Flushing	Some types of GEP-NENs can produce hormones that cause flushing, a sudden redness, and warmth in the face and neck.

GEP-NEN, gastroenteropancretic neuroendocrine neoplasm.

**Table 5 diagnostics-13-02741-t005:** Diagnostic tools and new techniques.

Modality/Diagnostic Tool	Characteristics	Limitations
Contrast-enhanced CT	- First tool of choice in most cases for diagnosis, staging, and follow-up - Broadly available - Good sensitivity	- Radiation exposure - Variable specificity - Can miss small pancreatic or gastrointestinal NENs
Contrast-enhanced MRI	- Often the modality of choice or complementary to CT - No radiation exposure	- (Relative) contraindications, e.g., metallic implants - Less available than CT
68Gallium-DOTA-TOC/-NOC/-TATE PET	- Tool of choice for well-differentiated NENs - Very good sensitivity and specificity - Sensitive in lymph node staging and detection of other metastases	- Not broadly available - Costs
Endoscopic ultrasound (EUS)	- Diagnostic tool for gastric, duodenal, rectal, and pancreatic NENs - Detection of small tumors - Enables simultaneous histological evaluation	- Invasive procedure - Depends on observers’ skills
PET/MRI	- Combination of SSTR-PET and MRI, especially DWI - No radiation exposure	- Costs - Only available in few centers - Prognostic factures still under investigation
Radiomics	- Conversion of quantitative image features into datasets - Available for CT and MRI - Encouraging results in first studies	- Not suitable for clinical routine - Further investigations necessary

CT, computed tomography; NEN, neuroendocrine neoplasm; MRI, magnetic resonance imaging; PET, positron emission tomography; SSTR, somatostatin receptor; DWI, diffusion-weighted imaging.

## Data Availability

Not applicable.

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
