# Peer review of "Gastroenteropancreatic Neuroendocrine Tumors—Current Status and Advances in Diagnostic Imaging"

_diagnostics, 2023, doi:10.3390/diagnostics13172741_

Round 1
Reviewer 1 Report
Dear Vogele and the team,
I had the pleasure in reding your manuscript "Gastroenteropancreatic Neuroendocrine Tumors – Current Status and Advances in Diagnostic Imaging". I have a suggestion,
1. Can you add a table of current diagnostic modalities and the new techniques available.
2. Can you please add a flow diagram for the approach of these lesions?
Otherwise the manuscript is well written.
Best wishes
Spellcheck
Author Response
Dear Reviewer,
Thank you very much for your help and your comments.
Point 1: Can you add a table of current diagnostic modalities and the new techniques available?
We added a table containig a summary of diagnostic tools and new techniques.
Point 2 Can you please add a flow diagram for the approach of these lesions?
We added a flow diagram for the approach of GEP-NEN (Fig 1)
On behalf of the authors,
Daniel Vogele
Reviewer 2 Report
To the Authors,
as follows my comments on the manuscript:
The paper is presenting a review of current imagistic diagnostic options for GEP-NEN. It is in my opinion adequately covering the subject, providing an overview of the different origins of GEP-NEN, the different clinical manifestations and prognosis according to size, localization and type, as well as their individualized diagnostic options.
Based on the current literature and data, the authors conclude that CT and MRI have high detection rates of GEP-NEN and should be involved in the primary workup, while nuclear medicine procedures with superior sensitivity can also provide important information. The diagnostic possibilities could be improved in the future, with the development of new techniques like radiomics.
The manuscript is in my opinion well structured. The cited references are relevant to the topic. The figures are clear and easy to understand.
I find the conclusions are consistent with the argumentation; they are in concordance with previous published data. The conclusions are supported by the listed citations.
The work is no novelty on the field but a nice and clear review of diagnostic steps for the work-up of the heterogenous entity of GEP-NEN.
I suggest a minor revision of the language, the sentence: “The different types of GEP-NENs, typical appearance, symptoms and the corresponding imaging capabilities are presented in the following.” is written twice (lines 89-91 and 104-105). I suggest revising the text on line 49: “which may involve mesenteric vessels and fix intestinal loops” and on line 264: “due to their size and location and thus resemble similar to ductal adenocarcinoma.”
Author Response
Dear Reviewer,
Thank you very much for your help and your comments which improved our manuscript.
Point1: “The different types of GEP-NENs, typical appearance, symptoms and the corresponding imaging capabilities are presented in the following.” is written twice (lines 89-91 and 104-105).
Sentence in lines 89-91 has been deleted.
Point 2: line 49: “which may involve mesenteric vessels and fix intestinal loops”
Has been corrected.
Point 3: line 264: “due to their size and location and thus resemble similar to ductal adenocarcinoma.”
Has been corrected.
On behalf of the authors,
Daniel Vogele